# Alveolar Bone Remodeling with or without Collagen Filling of the Extraction Socket: A High-Resolution X-ray Tomography Animal Study

**DOI:** 10.3390/jcm11092493

**Published:** 2022-04-29

**Authors:** Ugo Covani, Enrica Giammarinaro, Daniele Panetta, Piero A. Salvadori, Saverio Cosola, Simone Marconcini

**Affiliations:** 1Istituto Stomatologico Toscano, Via Aurelia 335, 55041 Lido di Camaiore, Italy; covani@covani.it (U.C.); s.cosola@hotmail.it (S.C.); 2CNR Institute of Clinical Physiology (CNR-IFC), Via G. Moruzzi 1, 56124 Pisa, Italy; daniele.panetta@ifc.cnr.it (D.P.); salvador@ifc.cnr.it (P.A.S.)

**Keywords:** extraction socket, alveolar bone remodeling, micro-CT, animal study, bone grafting

## Abstract

The healing process of the tooth extraction socket often leads to significant resorption of the alveolar bone, eventually causing clinical difficulties for future implant-supported rehabilitations. The aim of the present animal study was to evaluate alveolar bone remodeling after tooth extraction in a rabbit model, either with or without the use of a plain collagen plug inside the socket, by means of micro-computed tomography. The study included the micro-tomography analysis of 36 rabbits’ incisor extraction sockets, either left empty or filled with a collagen plug. All animals were euthanized in a staggered manner, in order to address molecular, histologic, and radiographic analyses at different time-points, up to 90 days after surgery. The three-dimensional evaluation was carried out using micro-computed tomography technology on excised bone blocks including the alveolus and the contralateral bone. Both linear and volumetric measures were recorded: the percentage of bone volume change (ΔBV) within the region of interest was considered the primary endpoint of the study. The micro-CT analysis revealed mean volumetric changes of −58.1% ± from baseline to 3 months for the control group, and almost no bone loss for the test group, −4.6%. The sockets treated with the collagen plug showed significantly less dimensional resorption, while the natural-healing group showed an evident collapse of the alveolar bone three months after extraction surgery.

## 1. Introduction

In 1963, Atwood claimed that:

“*The uniform and continuous nature of the zone of external resorption of the residual ridge suggests the presence of an external stimulus to osteoclastic resorption from an undetermined origin. It might be a dental prosthesis, increased mucosal vascularity, a constricting mucoperiosteum, muscle action, the trauma of mastication, or some other factor yet unknown*” [1].

Today, the external stimulus Atwood was talking about is still disputed, but we know for sure that tooth extraction results in the tridimensional remodeling of the alveolar bone. Up to 50% of the bone volume is reported to be lost during the initial period of repair [2]. Systematic reviews have demonstrated that the alveolar socket undergoes an average horizontal shrinkage between 0.9 and 3.8 mm and an average vertical reduction of 1.24 mm, within 3 to 7 months after tooth extraction [3,4]. Overall, these changes are more pronounced at the buccal plate [5].

The study of alveolar sockets in animal models can provide valuable information about the healing events that occur between tooth extraction and complete bone regeneration [6]. The volumetric analysis of the ridge (in horizontal and vertical planes), as well as its micro-architecture, is very useful to determine the efficacy of alveolar ridge preservation (ARP) strategies [7]. Those techniques can attenuate the resorption process compared to natural healing, although they fail to preserve the entire volume of the socket [8]. Thus, no specific technique is judged as superior, based on the current evidence [9,10].

The variability in final bone quality depends on the resorption rate of the graft material and its osteoinductive properties, as well as on the wound size and nutrition supply [11]. Collagen plugs resorb within 15 days and are very porous and allow fast vascularization of the socket [12].

Until recently, histologic analyses were the standard for assessing alveolar bone structures. However, they have limitations with respect to bone microarchitecture evaluation, because structural measures are derived from stereologic analysis of a few 2D sections [13].

Microcomputed tomography (or high-resolution X-ray tomography, micro-CT) has many advantages: (i) it allows 3D measurement of the trabecular pattern of the bone, (ii) you can analyze a large volume, (iii) measurements are faster than histomorphometric analyses, and (iv) the assessment is nondestructive, thus, samples can be used for further assays [14].

The present study featured the three-dimensional evaluation of bone healing after tooth extraction in rabbits, with and without post-extraction collagen filling of the socket. The study is part of a larger investigation that included 36 rabbits and both histologic and molecular analysis of the extraction sockets and the results have already been published. The study hypotheses were that (1) micro-CT analysis would demonstrate significant reduction in alveolar bone dimensions and volume after extraction over a 90-day period, and (2) micro-CT analysis would demonstrate significantly less bone resorption in sockets filled with a collagen plug. The main aim was to assess the volumetric change of the socket according to the treatment strategy.

## 2. Materials and Methods

### 2.1. Study Design

The present protocol was written in accordance with the international ARRIVE guidelines that are meant to improve the quality of research using animals. The randomization sequence was derived by a statistical software. No a priori criteria to exclude animals were applied. The randomization sequence was simple, thus eventual confounders were not controlled.

A single tooth extraction was performed on each rabbit, and then the socket was allocated to two possible management procedures: in the first group, soft tissues were closed with tension-free sutures but the socket was left empty (control); in the second group, the extraction socket was filled with a collagen plug and no suture was applied (test). In the second step of the investigation, the animals were divided into six groups according to the time between the tooth extraction and the moment of analysis.

### 2.2. Sample Size Calculation

The original protocol included 36 rabbits; the sample size being chosen according to the ethical principle of minimum sacrifice but sufficient power [15]. The law of diminishing return was applied as there were no previous information about effect size. 

According to this method, the value “E” (the degree of freedom of analysis of variance) should lie between 10 and 20. E was measured by the following formula:E = Total number of animals − Total number of groups(timepoints × treatment groups)

In this case, E for 36 animals was
E = (36) − (6 × 2) = 24

Even if E for 36 animals was slightly higher than 20, it was considered correct when taking into account 10% possible animal death.

### 2.3. Animal Management

Study approval was obtained from the Ethical Commission for Animal Welfare, Pisa, Italy (IRB 0035123/2017). A total of thirty-six white adult male New Zealand rabbits with an average body weight of 2 kg were housed at the Veterinary Department of the University of Pisa. The animals’ behavioral and health conditions (aeration, food, water administration, any symptoms) were checked throughout the entire study period. All animals were pre-medicated with an intramuscular injection of 0.2 mL meloxicam (Metacam, Boehringer Ingelheim, Ingelheim am Rhein, Germany, 0.5 mg/kg). The anesthesia was achieved with 0.8 mL of intramuscular alphaxolone (Alfaxan; Jurox UK, Worcestershire, UK, 10 mg/mL) and infiltration of local anesthetic (Xylocain Dental adrenalin, Astrazencea, Milano, Italy, 20 mg/mL + 12.5 mg/mL) was performed. Systemic antibiotics (Baytril, Bayer S.p.A., Milano, Italy, 25 mg/mL) were administered for five days for prophylaxis.

### 2.4. Surgical Phase

All surgeries were performed in an animal operating theater under general anesthesia. The lower right or left incisor was luxated with a small elevator (iM3) and then carefully extracted with rabbit-specific extraction forceps (iM3). The integrity of the socket was probed and then the site was assigned to either the control group (natural healing with tension-free suture) or the test group (the socket was filled with native heterologous collagen type 1 of lyophilized and sterile bovine origin (ACE) plugs and no suture was applied).

### 2.5. Terminal Procedure

The thirty-six animals were euthanized in groups of six at 2, 7, 15, 30, 60, and 90 days after surgery. Respiratory arrest was induced with an intravenous injection of a 20% solution of pentobarbital.

### 2.6. Sample Preparation

Block resections of the anterior mandible were performed by means of an oscillating autopsy saw. Each bone block included the extraction socket and the contra-lateral one in order to preserve an internal control for linear and volumetric measurements. Bone blocks were preserved in formalin until analysis.

### 2.7. High-Resolution X-ray Tomography

To test the course of bone healing, rabbits were euthanized at two days, one week, two weeks, four weeks, eight weeks, and twelve weeks after tooth extraction. The blocks containing the anterior mandible segments were scanned with X-ray micro-CT scanner IRIS (Inviscan Sas, Strasbourg, France) at an isotropic voxel size of 59 μm. The acquisition used X-ray beam quality of 80 kV with a tube current of l mA, over a total scan time of 112 s per sample. Reconstructions were performed with cone-beam filtered back projection (FBP), with standard reconstruction kernel, after raw data pre-correction for beam hardening artifacts.

The threshold for the bone-only label of each segmentation was chosen approximately in between the mean CT numbers in bone marrow and bone, i.e., at ca. 500 HU.

Images of each bone specimen were reoriented and displayed from coronal to apical level using multiple-planar reformation (MPR), which provided axial cross sections of the inner structure of the samples. The image reorientation was conducted in such a way to allow the positioning of the two sockets parallel to the floor, using the incisal margin of the contra-lateral incisor as a reference (experimental control). The intersection of horizontal and vertical lines determined the position of the socket centroid for each image. The horizontal line defined the extremities from the outer portion of each side. The distance between the centroid to the external perimeter of the socket represented the linear measurement to determine bone change.

A cylindrical region of interest was determined by segmenting the bone located from the coronal to the 10th mm apical third. Reference points for linear measurements of the bucco-lingual dimension were set at 1-mm intervals starting from 1 mm apical to the cemento-enamel junction (CEJ) to 10 mm apically.

### 2.8. Linear and Volumetric Measures

Linear measurements included the linear distance from the CEJ to the ridge crest measured by drawing a horizontal line between the CEJs of teeth adjacent to the extraction site and recording the vertical distance from that line to the bone crest in millimeters and the surface area of the selected volume.

The tridimensional outcome measurements included bone volume (BV) and bone volume change (ΔBV) in percentage which was considered the primary outcome measure. A significant clinical treatment effect “d” would have required 25% less bone resorption in the test group. To this purpose, the bone component of each sample, starting from the coronal margin and down to a depth of 10 mm along the coronal direction, was segmented. The resulting labeled volumes have been virtually filled to measure their total volume. In order to cope with the non-planar shape of the cervical line, a best-fit closed surface was selected to virtually fill up the socket; the same surface was used on the contralateral socket to cut out the tooth crown from the labeled volume, hence obtaining comparable measurements between the two sockets. The software 3DSlicer (v. 4.11) was used for all the volumetric measurements [16].

### 2.9. Statistical Analysis

An exploratory study was performed on the data relating to the variables studied. Statistical analyses were performed with R version 4.0.4 (15 February 2021)—“Lost Library Book” Copyright © 2021 The R Foundation for Statistical Computing Platform. The Brunner–Langer approach (mixed nonparametric method for longitudinal analysis) was employed. The F1-LD-F1 function (nparLD package) performs several tests for the relative treatment effects with global or patterned alternatives for the non-parametric design with one sub-plot factor (treatment group) and one whole-plot factor (time). This function retrieves the Wald-type statistic (WTS), the ANOVA-type statistic (ATS), and the modified ANOVA type statistic with Box approximation for testing group and time effects, and interaction. Findings were considered statistically significant at 0.05.

## 3. Results

All extraction sites healed uneventfully with no complications requiring surgical intervention or administration of additional medications. No animals or bone samples were excluded from the analysis.

The summary descriptive statistics showed no morphometric differences between groups or animals at baseline, thus corroborating the initial assumption of similarity between subjects. The mean initial volume of interest for the test group was 213 ± 0.10 mm^3^ and 180 ± 0.70 mm^3^ for the control group.

The longitudinal analysis showed a substantial reduction in the median alveolar vestibular bone height, with greatest resorption at the control extraction sites (median 3.0 mm against 0.4 mm of the test group).

The average volumetric percentage remodeling (mm^3^) was significantly different between groups (*p* value = 0.02), with the control group (no collagen) showing greater bone resorption at the extraction site in all directions (−58.0%), and the test group showing essential preservation of the initial bone profile (−4.6%), at a 90 day evaluation (Table 1).

After 90 days, the position of the crestal bone at the center of the extraction socket appeared to be 2.0 mm below and 2.00 mm inward with respect to its original position. Subtracted images of the area of the extraction sockets revealed mean volumetric changes of −58.1% ± 10% when comparing baseline to 3 months for the control group. Virtually, no bone loss was observed for the test group.

The greatest difference in bone volume change between groups was observed from day 14 to 30, with the first two weeks showing almost the same evolution and the rest of the period denoting a marked gap between groups.

The control group showed massive tridimensional collapse of the alveolar walls at three months, and this qualitative macroscopic finding was confirmed with the statistical analysis for non-parametric data.

The trabecular pattern of the new-forming bone showed growth with changes in shape, thickness, and porosity in the times evaluated. These changes represented an increase in trabeculae thickness and a reduction in open spaces. Three-dimensional images revealed the extraction sockets at the initial period completely void with a lack of hyper-dense areas, while at 7 days, hyper-dense areas were evidenced compatible with the beginning of bone formation at the most apical region of the socket. At 30 days, the hyper-dense areas evidenced advanced bone formation with the trabecular bone extending centripetally, while at 90 days, larger hyper-dense areas were observed in the entire socket, evidencing that the extraction socket was completely healed.

The comparison of the ‘representative’ microcomputed tomographic images of each group evidenced the difference in bone crest architecture between the groups after 3 months of treatment. The better preservation of buccal bone crest in the test group can be noticed (Figure 1 and Figure 2).

## 4. Discussion

The present micro-CT study reported the outcomes of a 90-day healing period of a rabbit extraction socket model. The sockets were either filled with a collagen plug or not. The sockets treated with the collagen plug showed significantly less tridimensional resorption, while the natural healing group showed evident collapse of the entire socket. This finding might be related to the collagen plug capacity to act as an anchorage for the provisional ECM whilst the early healing process was taking place [17].

The biophysical properties of the ECM, such as stiffness and topography, directly influence cellular functions, especially in the context of wound healing [18]. In many cell types, including fibroblasts, loss of substrate adhesion results in anoikis (anchorage-dependent apoptosis) [19]. Different models have suggested that physical anchorage of the extracellular matrix (ECM), eventually via the aid of biomaterials, promotes granulation tissue survival [20].

Furthermore, the collagen plug might have helped in creating a tent effect for the periosteum, which is positively stimulated by tensile strain. When tensile strain is applied to the periosteum, mesenchymal stem cells of periosteal origin differentiate into osteogenic cells and induce early sub-periosteal callus formation [21]. When the periosteal architecture is maintained and the periosteum tented away, bone formation can occur, even if limited by a line joining the crests of the socket. The collagen plug might be sufficient to attenuate excessive granulation tissue contractility during the early stages of wound healing [22]. Myofibroblasts are the differentiated line of fibroblasts and they are responsible for wound contraction [23]; they activate upon ECM change in stiffness via specialized mechano-sensors. It is plausible that excessive contraction at the most coronal portion of the alveolus might induce the collapse of the underlying lightest bone wall, the vestibular one.

Some authors have argued that alveolar ridge preservation might cause hindrance to normal socket healing [24], shows no benefit, and particles of different grafting materials may remain in the extraction socket for an indefinite time, leading to non-predictable autologous bone substitution [25,26].

One major factor during bone healing is sufficient vascularization and oxygen tension, which provide the necessary cells and nutrients [27,28]; disturbance or lack of vascularization may result in delayed healing and/or compromised integration of the grafting material [29]. Beside its scaffold effect, collagen degradability makes it the perfect candidate as it would enable ECM remodeling which is highly important for cellular proliferation and migration [30].

In addition, it should be emphasized that the radiographic measurements of alveolar ridge dimensions may not reflect the true ridge dimensions since it is not possible to differentiate the new bone formation from the remaining graft particles on radiographs. Hence, the outcomes of radiographic studies should be interpreted with caution.

However, considering the number of studies published on the efficacy of alveolar ridge preservation procedures, the question today is no longer to identify whether ARP procedures can prevent post-extraction bone loss, but instead to determine in which clinical scenarios these procedures are most beneficial [31]. As long as the tooth extraction is simple, the esthetic is not that demanding, the residual bone volume is fair, and the eventual left defect is a space-keeping one, a collagen plug may be sufficient to prevent excessive bone loss.

The present study has some limits that must be emphasized. Given the pre-clinical design, its results may not be generalized to human subjects; rabbits show continuous growth of the incisors, which also could have disturbed the current study’s most recent follow-up evaluation; the longitudinal design of the study protocol imposed the sacrifice of different animals representing different moments of the follow-up, thus assuming that all the rabbits were perfectly identical at the beginning of the study.

## 5. Conclusions

The present micro-CT animal study demonstrated the importance of preserving the position of hard and soft tissues after single tooth extraction by means of a simple collagen plug filling the empty socket. At three months, the control socket had reduced in volume by 50%, while no bone loss was observed for the test group. The collagen plug, within an intact alveolus, might be sufficient to prevent extensive tridimensional collapse of the alveolar bone, thus favoring future rehabilitation. Further human studies, with long follow-up periods, are needed to prove the clinical significance of the present findings.

## Figures and Tables

**Figure 1 jcm-11-02493-f001:**
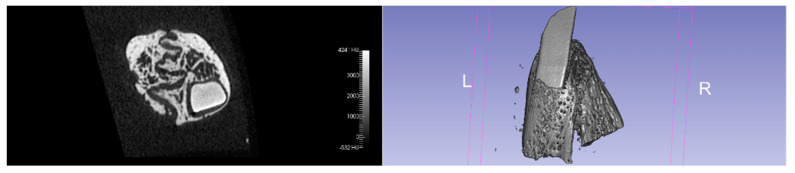
Coronal section and tridimensional reconstruction of the control group extractive socket at a 90-day evaluation.

**Figure 2 jcm-11-02493-f002:**
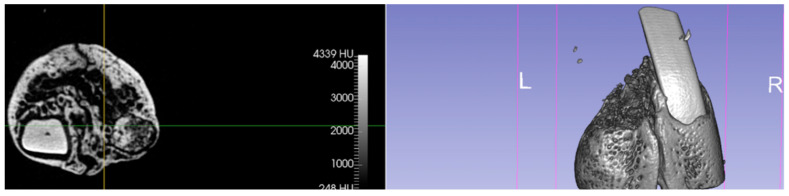
Coronal section and tridimensional reconstruction of the test group extractive socket at a 90-day evaluation.

**Table 1 jcm-11-02493-t001:** Percentage volumetric bone remodeling in both groups and WTS *p*-value.

Volumetric Bone Change %	
Test group	
Baseline to 14 days	Baseline to 90 days
−10.7%	−4.6%
WTS *p*-value 0.02995674.	
Control group	
Baseline to 14 days	Baseline to 90 days
−41%	−58.1%

## Data Availability

The data that support the findings of this study are available from the corresponding author upon reasonable request.

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
