# Peer review of "Alveolar Bone Remodeling with or without Collagen Filling of the Extraction Socket: A High-Resolution X-ray Tomography Animal Study"

_jcm, 2022, doi:10.3390/jcm11092493_

Round 1

Reviewer 1 Report

Dear Authors first of all congratulations on the study, It is difficult to investigate with animaos nowadays.

I would like to ask you something about your study and give you some points of improvements:

In your study design you should write your objectives/aims and give some conclusions on your work. This is very important and I miss that in your article.

I think you should rewrite the first paragraph on your discussion, you are expalining and not discussing, si that paragraph should be In the introduction.

thank you

Author Response

Rebuttal jcm-1692323

Dear Editor, the text has been checked and modified in order to diminish the repetition rate.

Reviewer 1

Dear Authors first of all congratulations on the study, It is difficult to investigate with animals nowadays.

Dear reviewer, thank you for your interest in our study.

I would like to ask you something about your study and give you some points of improvements:

In your study design you should write your objectives/aims and give some conclusions on your work. This is very important and I miss that in your article.

Dear reviewer, thank you for your precious suggestion. The study design section has been improved adding the aims of the study. The conclusion section has been added as well.

I think you should rewrite the first paragraph on your discussion, you are explaining and not discussing, si that paragraph should be In the introduction.

Dear reviewer, thank you for you help. The discussion incipit has now been slightly modified according to your suggestion.

Reviewer 2 Report

01

 In 1963, Atwood claimed that:

The uniform and continuous nature of the zone of external resorption of the residual ridge suggests the presence of an external stimulus to osteoclastic resorption from an un-determined origin. It might be a dental prosthesis, increased mucosal vascularity, a constricting mucoperiosteum, muscle action, the trauma of mastication, or some other factor yet unknown [1].

If this passage of text is a verbatim copy of the article of Atwood (as the authors used the verb “claimed”), then this passage should be between quotation marks (“”). Maybe also in italics.

02

There are some sentences in the text without reference to a previous study (or studies) in order to give evidence to their statements. Without references, these statements would be mere assumptions or allegations by the author of the thesis. Therefore, each of the following sentences need at least one reference to back up their statement:

“Overall, these changes are more pronounced at the buccal plate.”

“The degree of change in bone quality depends on the resorption rate of the graft material and its ability to encourage bone formation, as well as on the wound size and nutrition supply.”

“Collagen plugs resorb within 15 days and are very porous and allow fast vascularization of the socket.”

“This finding might be related to the collagen plug capacity to act as an anchorage for the provisional ECM whilst the early healing process was taking place.”

“In fact, the biophysical properties of the ECM, such as stiffness and topography, directly influence cellular functions, especially in the context of wound healing.”

“In many cell types, including fibroblasts, loss of substrate adhesion results in anoikis (anchorage-dependent apoptosis).”

“Thus, for the early healing tissue, having ECM anchorage might promote faster and more sound repair.”

03

“The sample size was chosen according to the ethical principle of minimum sacrifice but sufficient power related to the molecular investigation [12].”

Please add a sub-section called “Sample Size Calculation” to the Materials and Methods, and describe in details the power analysis.

04

“The sample size was chosen according to the ethical principle of minimum sacrifice but sufficient power related to the molecular investigation [12].”

The authors mentioned the samples size was chosen “according to the ethical principle of minimum sacrifice”. Yet, there was only one tooth extraction per rabbit. This does not fulfill the ethical principle of “minimum” sacrifice at all. This is actually “doing the minimum in each rabbit”.

05

“unmatched Wilcoxon signed ranks where used to measure differences in all samples.”

How many groups were being compared when Wilcoxon signed ranks was used? In other words, the Wilcoxon signed ranks test was used to compare the results of how many groups at the same time?

There are two groups, investigated in different time points (2, 7, 15, 30, 60, and 90 days).

06

The authors used ANOVA for the comparison of multiple groups, while the groups had only 6 specimens. If the number of samples per group is low, the analysis calls for a non-parametric test regardless of the normality, as the normal distribution cannot properly be verified.

07

“The average volumetric percentage remodeling (mm3) was significantly different be-tween groups (p value= 0.02), with (…)”

That is the only p value that I was able to find in the text, while the authors mentioned that they tested more than only one comparison in the Materials and Methods.

Please present a table with all the comparisons tested in the study, with all the p values, and which statistical test was performed for every p value given.

08

The authors need to draw a conclusion after the discussion.

Author Response

Reviewer 2

Dear reviewer, thank you for help in improving the quality of our manuscript. You can find our point-per-point rebuttal below:

01

 In 1963, Atwood claimed that:

The uniform and continuous nature of the zone of external resorption of the residual ridge suggests the presence of an external stimulus to osteoclastic resorption from an un-determined origin. It might be a dental prosthesis, increased mucosal vascularity, a constricting mucoperiosteum, muscle action, the trauma of mastication, or some other factor yet unknown [1].

If this passage of text is a verbatim copy of the article of Atwood (as the authors used the verb “claimed”), then this passage should be between quotation marks (“”). Maybe also in italics.

Dear reviewer, thank you for your suggestion. It is, in fact, a verbatim copy of the Atwood article. We have changed the format, it is now in italics and between quotation marks.

02

There are some sentences in the text without reference to a previous study (or studies) in order to give evidence to their statements. Without references, these statements would be mere assumptions or allegations by the author of the thesis. Therefore, each of the following sentences need at least one reference to back up their statement:

“Overall, these changes are more pronounced at the buccal plate.”

“The degree of change in bone quality depends on the resorption rate of the graft material and its ability to encourage bone formation, as well as on the wound size and nutrition supply.”

“Collagen plugs resorb within 15 days and are very porous and allow fast vascularization of the socket.”

“This finding might be related to the collagen plug capacity to act as an anchorage for the provisional ECM whilst the early healing process was taking place.”

“In fact, the biophysical properties of the ECM, such as stiffness and topography, directly influence cellular functions, especially in the context of wound healing.”

“In many cell types, including fibroblasts, loss of substrate adhesion results in anoikis (anchorage-dependent apoptosis).”

“Thus, for the early healing tissue, having ECM anchorage might promote faster and more sound repair.”

Dear reviewer, thank you for you important insight. Each sentence you mentioned has now been paired to its original reference. Please find the highlighted referenced in the modified manuscript.

03

“The sample size was chosen according to the ethical principle of minimum sacrifice but sufficient power related to the molecular investigation [12].”

Please add a sub-section called “Sample Size Calculation” to the Materials and Methods, and describe in details the power analysis.

Dear reviewer, thank you for you help, the sample size calculation section has been included as you suggested.

04

“The sample size was chosen according to the ethical principle of minimum sacrifice but sufficient power related to the molecular investigation [12].”

The authors mentioned the samples size was chosen “according to the ethical principle of minimum sacrifice”. Yet, there was only one tooth extraction per rabbit. This does not fulfill the ethical principle of “minimum” sacrifice at all. This is actually “doing the minimum in each rabbit”.

Dear reviewer, thank you for your precious comment. We understand your point of view. The number of tooth extractions per rabbit was defined depending on several factors, considering that both molecular and micro-CT analysis were to be computed for each sample. The rabbit model for tooth extraction classically comes with one tooth extraction per animal; this choice was also corroborated by both the leading veterinarian and the animal ethical committee, as they were in charge of ensuring animals well-being and ability to feed themselves unassisted after surgery.

05

“unmatched Wilcoxon signed ranks where used to measure differences in all samples.”

How many groups were being compared when Wilcoxon signed ranks was used? In other words, the Wilcoxon signed ranks test was used to compare the results of how many groups at the same time?

There are two groups, investigated in different time points (2, 7, 15, 30, 60, and 90 days).

 Dear reviewer, thank you for your question. The sentence was actually miswritten and now it has been revised. The F1-LD-F1 function (nparLD package) was used for all analysis, as the low sample size required non parametric longitudinal analysis.

06

The authors used ANOVA for the comparison of multiple groups, while the groups had only 6 specimens. If the number of samples per group is low, the analysis calls for a non-parametric test regardless of the normality, as the normal distribution cannot properly be verified.

 Dear reviewer, thank you for your precious question. The mentioned ANOVA is the one modified by Brunner in 2002 and it is meant for longitudinal data in nonparametric factorial settings. The nparLD function retrieves both WTS and ATS. In this analysis, both WTS and ATS yield highly statistically significant p values. However, we have corrected the text, making it clear that WTS and ATS are both displayed automatically when running the function. 

07

“The average volumetric percentage remodeling (mm3) was significantly different between groups (p value= 0.02), with (…)”

That is the only p value that I was able to find in the text, while the authors mentioned that they tested more than only one comparison in the Materials and Methods.

Please present a table with all the comparisons tested in the study, with all the p values, and which statistical test was performed for every p value given.

Dear reviewer, thank you for your precious contribution. The mention to statistical significance with regard to the volumetric percentage only was due to the fact that this was the primary outcome of the study. The other measurements were qualitative (trabecular pattern), linear, or showing a macroscopic difference between the two groups. Linear measurements were taken mainly to build the VOI. We have now added the requested table.

08

The authors need to draw a conclusion after the discussion.

Dear reviewer, thank you for your suggestion. Please find the new section in the text. 

Round 2

Reviewer 1 Report

Dear author, I believe this article has been improved since the suggestions I made and I think is better now but I still see something’s that should be changed:

The conclusions are still quite weak according to your study, and there are still somethings that I should write down like that there should be more studies in order to know if the collagen  is good or not. 

Author Response

Dear reviewer, 

thank you for your precious help in improving the quality of our manuscript.

Please find the revised conclusion at the end of the study.

Regards

Reviewer 2 Report

The manuscript now seems to be suitable for publication.

Author Response

Dear reviewer, thank you for appreciating our effort in improving the manuscript and thank you for your precious help in doing it.

Kind regards